# Pro-Osteogenic and Anti-Inflammatory Synergistic Effect of Orthosilicic Acid, Vitamin K2, Curcumin, Polydatin and Quercetin Combination in Young and Senescent Bone Marrow-Derived Mesenchymal Stromal Cells

**DOI:** 10.3390/ijms24108820

**Published:** 2023-05-16

**Authors:** Chiara Giordani, Giulia Matacchione, Angelica Giuliani, Debora Valli, Emanuele Salvatore Scarpa, Antonella Antonelli, Jacopo Sabbatinelli, Gilberta Giacchetti, Sofia Sabatelli, Fabiola Olivieri, Maria Rita Rippo

**Affiliations:** 1Department of Clinical and Molecular Sciences, DISCLIMO, Università Politecnica delle Marche, 60126 Ancona, Italy; 2Department of Biomolecular Sciences, University of Urbino Carlo Bo, 61029 Urbino, Italy; 3Clinic of Endocrinology and Metabolic Diseases, Department of Clinical and Molecular Sciences, Università Politecnica delle Marche, 60126 Ancona, Italy; 4Clinic of Laboratory and Precision Medicine, IRCCS Istituto Nazionale di Ricovero e Cura per Anziani, 60121 Ancona, Italy

**Keywords:** mesenchymal stromal cells, senescence, natural compounds, osteogenesis, inflammaging, osteoporosis

## Abstract

During aging, bone marrow mesenchymal stromal cells (MSCs)—the precursors of osteoblasts—undergo cellular senescence, losing their osteogenic potential and acquiring a pro-inflammatory secretory phenotype. These dysfunctions cause bone loss and lead to osteoporosis. Prevention and intervention at an early stage of bone loss are important, and naturally active compounds could represent a valid help in addition to diet. Here, we tested the hypothesis that the combination of two pro-osteogenic factors, namely orthosilicic acid (OA) and vitamin K2 (VK2), and three other anti-inflammatory compounds, namely curcumin (CUR), polydatin (PD) and quercetin (QCT)—that mirror the nutraceutical BlastiMin Complex^®^ (Mivell, Italy)—would be effective in promoting MSC osteogenesis, even of replicative senescent cells (sMSCs), and inhibiting their pro-inflammatory phenotype in vitro. Results showed that when used at non-cytotoxic doses, (i) the association of OA and VK2 promoted MSC differentiation into osteoblasts, even when cultured without other pro-differentiating factors; and (ii) CUR, PD and QCT exerted an anti-inflammatory effect on sMSCs, and also synergized with OA and VK2 in promoting the expression of the pivotal osteogenic marker ALP in these cells. Overall, these data suggest a potential role of using a combination of all of these natural compounds as a supplement to prevent or control the progression of age-related osteoporosis.

## 1. Introduction

Osteoporosis is a chronic skeletal disorder characterized by compromised bone strength that predisposes one to the risk of fractures, which in light of the demographic trend, is becoming an increasing concern for health professionals [1]. According to the International Osteoporosis Foundation (IOS, https://www.osteoporosis.foundation/facts-statistics/epidemiology-of-osteoporosis-and-fragility-fractures (accessed on 27 April 2023)), 21.2% of women and 6.3% of men over the age of 50 years will experience an osteoporotic fracture during their lifetime. There are many risk factors for the pathogenesis of osteoporosis. Some are modifiable lifestyle factors, such as nutrition and physical activities, medications, and nicotine or alcohol abuse [2], while others are unmodifiable, i.e., physiological hormone changes after menopause which directly affects bone mineral density (BMD) [3], and aging [4]. Therefore, women are more at risk of developing osteoporosis.

The skeleton is a metabolic active tissue, whose regulation mainly depends on a tight balance between bone resorption (osteoclasts) and bone deposition (osteoblasts) [5]. Osteoblasts differentiate from bone marrow mesenchymal stromal cells (MSCs) and are the leading cells of bone deposition, as they secrete osteocalcin, alkaline phosphatase and type I collagen [6,7]. This fine balance starts failing after the third decade of life when bone resorption increases due to steroid hormones estrogens and testosterone decrease [8], leading to a reduced BMD [9]. Furthermore, with aging, a decrease in the osteoprogenitor pool occurs [10] with a shift from osteoblastogenesis to adipogenesis, which ultimately leads to bone marrow fat accumulation [11] and the accumulation of senescent cells in the bone microenvironment [12]. Senescent bone cells acquire distinctive phenotypic and metabolic alterations, named senescence-associated secretory phenotype (SASP) with a pro-inflammatory activity (inflammaging), which has been hypothesized to be the leading cause of tissue dysfunction and bone loss [8].

It is unclear whether senescent cells may trigger or intensify postmenopausal osteoporosis. Preliminary studies in ovariectomized mice show that acute estrogen deficiency does not increase the number of senescent cells. Similarly, postmenopausal women treated with estrogens showed senescence markers in bone biopsies [13]. Taken together, these data suggest that estrogens and senescent cells act independently on the pathogenesis of osteoporosis. Therefore, hormone replacement therapy alone might be insufficient to prevent osteoporosis.

Treating osteoporosis primarily means preventing fractures and using medicines to strengthen bones, such as calcium, vitamin D supplements and bisphosphonates, in addition to or instead of hormone replacement therapy [14]. Other studies highlight the possible role of senolytics or senomodulators in age-related osteoporosis treatment [15]. Many researchers focused on new, natural, less expensive compounds to prevent osteoporosis, possibly without side effects. A pro-osteogenic effect of orthosilicic acid, a more stable and bioavailable form of silica derived from plant-based foods (e.g., cereals) has recently been documented [16,17]. Carlisle et al. found that physiological doses of orthosilicic acid may help in calcium deposition [18], acting on the expression of alkaline phosphatase (ALP), collagen 1 (COL1) and osteocalcin (OCN) through bone morphogenetic protein 2 (BMP2)/Smad/Runt-related transcription factor 2 (RUNX2) signaling pathway in vitro [19]. Similarly, vitamin K2 has been associated with an increased BMD and bone strength [20]. Notably, vitamin K species (K1 phylloquinone and K2 menaquinone) have long been correlated with bone protective activities, even if the mechanism of action is still poorly understood. The key factor of bone resorption is nuclear factor κB (NF-κB) which induces osteoclasts formation and inhibits osteoblasts differentiation [21]. Yamaguchi and colleagues demonstrated that vitamin K2 down-regulates NF-κB through IκB mRNA expression, which ultimately inhibits osteoclasts formation in a γ-carboxylation manner [22]. Altogether, these data suggest a possible combined treatment for osteoporosis.

As for senotherapeutics, polyphenols have long been tested in animal models and human clinical trials to investigate their anti-inflammatory and antioxidant effects [23]. Curcumin, a widely studied nutraceutical derived from curcuma longa, and polydatin, a precursor of resveratrol, are considered powerful anti-inflammatory polyphenols that appear to restrain SASP [24,25]. Both are associated with inhibiting the receptor activator of NF-κB ligand (RANKL), one of the critical mediators of osteoclastogenesis [26,27]. Moreover, recent reports highlight other potential applications of polydatin to prevent bone loss [28] and curcumin in orthopedic research [29]. Quercetin is the most abundant flavonoid in fruits and vegetables, therefore it has been extensively studied to determine the effects of flavonoids [30]. Several clinical and preclinical studies show that quercetin enhances bone quality by reversing the differentiation inhibition of osteoblasts, exacerbated by SASP [31,32,33,34]. Along with the osteogenic effect mediated by the anti-NF-κB activity [35], it has been reported that quercetin exerts an effect directly on bone, increasing BMP signaling in rat and mouse bone marrow-derived MSCs [36,37].

To our knowledge, the pro-osteogenic and anti-inflammatory effects of orthosilicic acid and vitamin K2 (pro-osteogenic), curcumin, polydatin and quercetin (anti-inflammatory) combination on senescent bone marrow-derived MSCs has never been documented. Given the importance of senescence and SASP in the etiopathogenesis of osteoporosis, we decided to further investigate the potential effects of the above-mentioned natural compounds.

## 2. Results

### 2.1. Replicative Senescence in Human Bone Marrow Mesenchymal Stromal Cells

Senescence is associated with bone remodeling imbalance, and SASP strongly triggers bone resorption via a pro-inflammatory microenvironment [34]. Therefore, we decided to use MSCs in the early passages when they were at cPD < 4 (young, yMSC), but also when they reached the replicative senescence as a cellular model to study the beneficial effects of natural compounds. MSCs were considered senescent (sMSC) when they were at cPD > 14, (SA)-β-Gal activity was >60%, p16^ink4a^ and p21 expression was significantly higher and PCNA expression lower when compared to their level in yMSC. Telomere shortening was also evaluated to confirm the senescent status (Figure 1).

### 2.2. Orthosilicic Acid, Vitamin K2, Curcumin, Polydatin and Quercetin Effect on yMSC and sMSC Cell Viability

The cytotoxic effect of orthosilicic acid (OA), vitamin K2 (VK2), curcumin (CUR), polydatin (PD) and quercetin (QCT), alone and in the combination of OA and VK2 (OA+VK2), CUR, PD and QCT (C+P+Q) and the five compounds (MIX) was evaluated. Cell viability of yMSC and sMSC was analyzed after 24 h and 7 days of treatment by MTT assay (Figure 2). Further experiments were performed considering the concentration of compounds that gave at least >70% viability of treated cells (75 μM OA; 0.1 μM VK2; 1 μM CUR; 10 μM PD and 0.5 μM QCT).

### 2.3. Orthosilicic Acid and Vitamin K2 Have a Mineralizing Effect on yMSC

To test the possible pro-osteogenic efficacy of OA and VK2 on yMSC, we first characterised bone marrow mesenchymal stromal cells’ ability to differentiate into osteoblasts in vitro, by analysing the main osteogenic markers such as transcription factor Runt-related transcription factor 2 (RUNX2), alkaline phosphatase (ALP), collagen type 1 alpha 1 (COL1α1) and osteocalcin (OCN) using Alizarin Red staining [38] (Appendix A). yMSC were treated with OA, VK2 and their combination for 7 and 14 days without other pro-osteogenic factors (Figure 3). We observed that individual compounds were not effective at modulating the mRNA of differentiation markers, either after 7 or 14 days (Figure 3A,B), whereas their association (OA+VK2) successfully increased RUNX2, COL1α1 and OCN mRNA expression (Figure 3A), with a synergistic effect for RUNX2 and ALP after 7 days, and COL1α1 and OCN after 14 days (Figure 3B) (Table 1). Although ALP mRNA expression was not modulated either after 7 or 14 days, OA, VK2 and their association successfully increased its activity, which was tested after 7 days of treatment (Figure 3D). As one of the osteogenic promoters’ miRNAs, miR-98 expression was also evaluated [39,40]. Again, OA and VK2 alone slightly increased miR-98 expression, which was significantly up-regulated by their combination (Figure 3C). Protein expression analysis by Western blotting confirmed a significant up-regulation of Col1α1 after 7 days, and of OCN after 14 days of treatment with OA+VK2 (Figure 3E).

### 2.4. Curcumin, Polydatin and Quercetin Have Anti-Inflammatory Activity on sMSC

As inflammaging is associated with bone dysfunction [41] due to the accumulation of senescent cells, we verified the anti-inflammatory effect of a combination of curcumin (CUR, 1 μM), polydatin (PD, 10 μM) and quercetin (QCT, 0.5 μM) that had already been tested by our team [42] on sMSC. Following Matacchione et al.’s experimental settings [42], sMSC were treated with CUR, PD, QCT or their combination for 24 h. The analysis of the effect of single compounds showed that only PD was effective in downregulating the expression of pro-inflammatory markers, i.e., IL-1β and MCP-1 mRNAs (Figure 4A). However, their association (C+P+Q) further reduced IL-1β, MCP-1 mRNA and two miRs associated with cellular senescence and inflammaging, i.e., miR-21 and miR-146a expression [43] (Figure 4A). Moreover, PD was able to significantly reduce the release of MCP-1 in the conditioned medium, and the combination of natural molecules successfully decreased both IL-8 and MCP-1 release after 24 h treatment (Figure 4B). We also evaluated the expression of the activated p38 mitogen-activated protein kinases (MAPK) and the phosphorylated subunit p65 (NF-κB) as two of the key players of inflammation and bone resorption [21,43]. Western blotting and densitometric analysis showed that p38-MAPK and phosphorylated NF-κB were increased in sMSC compared to yMSC, and both efficiently reduced by C+P+Q (Figure 4C).

### 2.5. Natural Compound Combination (MIX) Exerts an Anti-Inflammatory and Osteogenic Effect on sMSC

Based on these initial data and cell viability assay (Figure 2), sMSC were treated with OA+VK2, C+P+Q and their combination (MIX) for 7 days to simultaneously appreciate the anti-inflammatory and pro-osteogenic effect on sMSC. Figure 5A shows that OA+VK2, C+P+Q and MIX exerted a striking effect on all cytokine mRNA expression compared to untreated control (sMSC) after 7 days of treatment, and MIX was also able to significantly decrease the expression of miR-146a. Moreover, the release of IL-8 and MCP-1 in the conditioned medium was successfully reduced after treatment with C+P+Q and MIX (Figure 5B). We found once again that the activation of p38-MAPK and the phosphorylation of NF-kB were increased in sMSC compared to yMSC (Figure 5D). Furthermore, C+P+Q and MIX successfully decreased both p38 MAPK and pNF-κB protein expression (Figure 5D). As for osteogenic markers, OA+VK2 treatment did not upregulate the pro-osteogenic markers except for ALP activity compared to untreated sMSC. Similarly, the MIX increased ALP mRNA expression via a synergism of OA+VK2 and C+P+Q (Table 2); accordingly, ALP activity was upregulated by both OA+VK2 and MIX (Figure 5C). Moreover, the MIX was efficient in raising Col1α1 mRNA expression. However, the MIX did not exert a valuable upregulating effect on miR-98 expression as the one observed with OA+VK2 (Figure 5A).

## 3. Discussion

Bone marrow mesenchymal stromal cells play a pivotal role in bone homeostasis. As precursors of osteoblasts, MSCs are directly involved in bone formation. During aging, they undergo stemness exhaustion and dysfunction as they more easily differentiate in fat cells. Furthermore, the low-grade chronic inflammation status of aging (inflammaging), a phenomenon that occurs due to the accumulation of senescent cells with a SASP phenotype, is also associated with osteoporosis [41]. In addition, estrogen deficiency due to menopause causes an increase in bone turnover, with an imbalance between bone resorption and bone formation. Overall, especially in women, a decrease in bone formation and marrow fat accumulation causes osteoporosis, with an increase in the risk of fractures that can be associated with significant morbidity and mortality. Management of skeletal health involves reducing modifiable risk factors (i.e., lifestyle and dietary changes), but also the use of pharmacological therapy for those patients at significant risk of osteoporosis and fractures [44]. Prevention and intervention at an early stage of disease are very important. This recommendation also applies to osteoporosis. Consumers are increasingly looking to natural health products to prevent or manage specific diseases. In the case of osteoporosis, these naturally active compounds could have their maximum effectiveness in the initial phase, osteopenia.

In this study, we wanted to assay the effect in MSCs of the combination of orthosilicic acid and vitamin K2, known for their beneficial effect on bone health and remodeling [16,33,45,46], and of curcumin, polydatin and quercetin for their synergism in the protection from inflammaging by decreasing the expression of inflammatory mediators produced by senescent cells [42]. We also tested the effect of a super combination of all these natural products, which here we called MIX, whose components are all included within BlastiMin Complex^®^ (Mivell, Italy).

We have demonstrated for the first time that OA and VK2 can synergize to promote differentiation of early passages (yMSC) into osteoblasts in the absence of a pro-osteogenic milieu. Indeed, their combination induced Runx2, the master and early marker of differentiation, and ALP at 7 days of treatment, and further Col1α1 and OCN, two later markers of the same process, at 14 days (Figure 6), reflecting the same timing of expression that is observed when cells are induced to differentiate with osteogenic medium.

Furthermore, we have shown that the association of curcumin, polydatin and quercetin, as already demonstrated for senescent endothelial cells [42], was able to modulate the SASP phenotype (Figure 6) of sMSC. This was observed by the reduction of IL-1β, MCP-1 mRNA expression and of the release of IL-8 and MCP-1 in the conditioned medium, the expression of the activated p38 mitogen-activated protein kinases (MAPK), and the phosphorylated subunit p65 (NF-κB) and, furthermore, the expression of two miR associated to cellular senescence and inflammaging, i.e., miR-21 and miR-146a. These experimental data are remarkable, since it has been demonstrated that sMSC can cause paracrine senescence of early passages MSC (yMSC) via inflammatory cytokines and the NF-κB pathway [47] (Figure 6). Therefore, it can be assumed that the association of these anti-inflammatory substances may play a role in the prevention of diseases characterized by low bone turnover and degeneration due to MSC senescence, such as osteoporosis. Overall, these results allowed us to test the effect of the association of all these natural compounds (MIX) on senescent MSC dysfunction, and in particular of the MIX double effect in reducing the inflammatory phenotype of MSCs on one hand and promoting their osteogenesis on the other. The results obtained showed that sMSCs, maintained in a non-pro-osteogenic medium, were stimulated by the MIX to up-regulate Col1α1 mRNA, alkaline phosphatase (ALP) activity and in a synergistic manner, ALP mRNA expression. Notably, although RUNX2 and OCN are not modulated upon treatment with natural compounds, the up-regulation of ALP underlines the effectiveness of the MIX in facilitating mineralization, presumably by increasing inorganic phosphate local rates and reducing the extracellular pyrophosphate concentration, which is an inhibitor of osteoblastogenesis [48]. In addition, our results showed that the presence of the pro-osteogenic compounds within the MIX did not negatively affect the anti-inflammatory action of the other combination constituents, since a reduction of all pro-inflammatory markers was maintained, NF-κB included (Figure 6). The effect of the MIX in the up-regulation of Col1α1 mRNA expression is noteworthy as well, since it showed the importance of using the five compounds combination to gain an effect in the differentiation. The ability of CUR, PD and QCT in reducing SASP may limit microenvironment alterations, thus promoting yMSC differentiation into osteoblasts. Given the role (i) of ALP in bone formation and as a good osteoporotic marker in post-menopause women [49], and (ii) of the negative effect that inflammation can play on osteogenesis, it could be surmised that the anti-inflammatory component of the MIX, together with the osteogenic ones, could promote the osteogenic process of MSCs.

## 4. Materials and Methods

### 4.1. Cell Culture

Human mesenchymal stem cells from the bone marrow (MSC) of a 30-year-old female donor were purchased from PromoCell (Heidelberg, Germany) and cultured in MSC growth medium 2 (MSC-GM2, PromoCell), with supplement mix (PromoCell) and 1% penicillin/streptomycin. Cells were seeded at a density of 5000/cm^2^ in T75 flasks (Corning Costar, Sigma Aldrich, St. Louis, MO, USA), at 37 °C in a humidified atmosphere with 5% CO_2_. The culture medium was changed every two days and cells were trypsinized when almost 80% confluent.

### 4.2. MSC In Vitro Differentiation

Osteogenic differentiation was achieved using mesenchymal stem cell osteogenic differentiation medium (PromoCell, Heidelberg, Germany). The medium was changed every 72 h; after 14 days, and cells were analyzed by Alizarin Red staining. Three photographs in 3 experiments were selected and run on the freely available imaging software ImageJ (version 1.52a, Stuttgart, Germany). Analysis was performed by threshold converting the 8-bit red–green–blue image into a binary image, consisting only of the pixels representing calcium deposits. The resulting percentage of the stained area was then compared to control images [50].

### 4.3. Replicative Senes Cence of Bone Marrow Mesenchymal Stem Cells

MSC achieved replicative senescence after several replicative passages, measured as population doubling (PDs) and calculated by the formula: (log10(F) − log10(I))/log10(2), where F is the number of the cells at the end of the passage and I is the number of seeded cells. Cumulative population doublings (cPD) were calculated as the sum of PD changes. MSCs were classified as replicative senescent cells (sMSC) as on senescence-associated (SA)-β-Galactosidase (SA-β-Gal) activity and p16^ink4a^, p21 and PCNA expression. The expression of pH-dependent SA-β-Gal activity was analyzed simultaneously in different MSCs passages using a senescence detection kit (BioVision Inc., Milpitas, CA, USA). Senescence was determined as a percentage by counting β-Gal-positive cells over at least 200 cells per well using light microscopy. P16^ink4a^ and p21 mRNA expression was evaluated by RT-qPCR; p16^ink4a^, p21 and PCNA protein expression were determined by Western blotting.

### 4.4. Natural Compounds

Orthosilicic acid was obtained by dissolving sodium metasilicate (Sigma Aldrich, Cat# 307815) in PBS. Vitamin K2 was purchased as menaquinone-7 (Sigma Aldrich, Cat# PHR2363) and dissolved in PEG-60 hydrogenated castor oil. Curcumin was purchased from Carlo Erba (Cat# A218580100), polydatin and quercetin from TCI (Cat# P1878 and P0042). These last compounds were dissolved in DMSO at a 0.1% final concentration of DMSO in all of the solutions. For all experiments, MSCs were seeded at a density of 5000/cm^2^ and treated after 24 h. The natural molecules analyzed in this study are part of the nutraceutical formulation BlastiMin Complex^®^ (Mivell, Italy. European Patent application: EP22216685.2).

### 4.5. Cell Viability Assay

Cell viability has been determined with MTT (3-(4,5-dimethylthiazol-2-yl)-2,5-diphenyltetrazolium bromide), a colorimetric assay that evaluates mitochondrial respiration efficiency. Cells were seeded in 96-well at a density of 5000 cells/cm^2^ before treatments with different doses of the selected compounds for 24 h and 7 days, to determine the maximum non-cytotoxic concentration. MTT (1 mg/mL) solution was added and incubated for 4 h. Insoluble formazan salts produced were solubilized by adding DMSO (100 μL), and then absorbance was measured at 540 nm using a microplate reader (NB-12-0035 Microplate Reader, NeoBiotech Co., Seoul, Republic of Korea). Data are expressed as a percentage of viability compared with untreated cells (control group considered 100% of viability) [42].

### 4.6. Cells Treatments

Based on the results of the viability assay, cells were treated: (i) for 7 and 14 days with orthosilicic acid (75 μM OA), vitamin K2 (0.1 μM VK2), or both (OA+VK2), or grown in either osteogenic differentiation medium as a positive control or MSC growth medium 2 as a negative control; (ii) for 24 h with curcumin (1 μM CUR), polydatin (10 μM PD), quercetin (0.5 μM QCT) and their combination (C+P+Q) or (iii) for 7 days with OA+VK2, C+P+Q and both (MIX). In these cases, untreated cells were grown in MSC growth medium 2 as a control. We considered the combinations as synergistic if the effect of the combination was greater than the sum of the effects of each compound acting separately.

### 4.7. RNA Isolation, mRNA Expression and miRNA Analysis by RT-qPCR

Total RNA was isolated with Norgen Biotek kit (#37500, Thorold, ON, Canada) according to the manufacturer’s instructions, and the total amount was determined with spectrophotometric quantification using Nanodrop ONE (NanoDrop Technologies, Wilmington, DE, USA). Total RNA (500 ng) was reverse transcribed using PrimeScriptTM RT Reagent Kit with gDNA eraser (TAKARA, Cat. RR047A) following the manufacturer’s instructions. RT-qPCR was performed in a Rotor-Gene Q (Qiagen) using TB Green™ Premix Ex Taq™ (Cat: RR420A). The following primers were all acquired from Merck Millipore (Darmstadt, Germany): IPO8 (FW: 5′-CGTTCCTCCTGAGACTCTGC-3′, RV: 5′-GAATGCCCACTGCATAGGTT-3′), Col1α1 (FW: 5′-CCAAATCTGTCTCCCCAGAA-3′, RV: 5′-TCAAAAACGAAGGGGAGATG-3′), ALP (FW: 5′-GAGAAGCCGGGACACAGTTC-3′, RV: 5′-CCTCCTCAACTGGGATGATGC-3′), RUNX2 (FW: 5′-GCGGTGCAAACTTTCTCCAG-3′, RV: 5′-TGCTTGCAGCCTTAAATGACTC-3′), OCN (FW: 5′-CTCACACTCCTCGCCCTATTG -3′, RV: 5′-GCTTGGACACAAAGGCTGCAC-3′), IL-1b (FW: 5′-AGATGATAAGCCCACTCTACAG-3′, RV: 5′-ACATTCAGCACAGGACTCTC-3′), IL-6 (FW: 5′-TGCAATAACCACCCCTGACC-3′, RV: 5′-GTGCCCATGCTACATTTGCC-3′), IL-8 (FW: 5′-GGACAAGAGCCAGGAAGAAA-3′, RV: 5′-CCTACAACAGACCCACACAATA-3′), MCP-1 (FW: 5′-GGCTGAGACTAACCCAGAAAAG-3′, RV: 5′-GGGTAGAAACTGTGGTTCAAGAG-3′), p16^INK4a^ (FW: 5′-GATCCAGGTGGGTAGAAGGTC-3′, RV: 5′-CCCCTGCAAACTTCGTCCT-3′), p21 (FW: 5′-CCATCCCTCCCCAGTTCATT-3′, RV: 5′-AAGACAACTACTCCCAGCCC-3′). mRNA quantification was assessed using the 2^−ΔΔCt^ method. Importin 8 was used as an endogenous control. MiRNAs expression was quantified by RT-qPCR using TaqMan miRNA assay (all from Thermo Fisher Scientific), according to the manufacturer’s protocol (miR-21 ID00397, miR-146a ID000468, miR-98 ID000577, RNU44 ID001094). Data were analyzed with Rotor Gene Q (Qiagen, Hilden, Germany) and standardized to RNU44. The 2^−ΔΔCt^ method was used to determine miRNA expression. 

### 4.8. DNA Extraction and Telomere Length Measurement

DNA was isolated using QIAamp DNA mini kit (Qiagen, Germany), according to the manufacturer’s instructions. DNA was stored at −80 °C until use. Telomere length was quantified with RT-qPCR by comparing the amount of the telomere amplification product (T) to that of a single-copy gene (S), which is *36B4* gene. The T/S ratio was then calculated to obtain the average telomere length [51]. The following primers acquired from Merck Millipore (Darmstadt, Germany) were used: TEL (FW:5′-GGTTTTTGAGGGTGAGGGTGAGGGTGAGGGTGAGGGT-3′, RV: 5′-TCCCGACTATCCCTATCCCTATCCCTATCCCTATCCTCA-3′).

### 4.9. Western Blot Analysis

Cell pellets were lysed in RIPA lysis buffer (150 mM NaCl, 10 mM Tris, pH 7.2, 0.1% SDS, 1.0% Triton X-100, 5 mM EDTA, pH 8.0), with protease inhibitor cocktail and phosphatase inhibitor PhosSTOPTM EASY pack (Roche Applied Science, Indianapolis, IN, USA) to extract total proteins. A Bradford assay and a microplate reader (NeoBiotech Co., Seoul, Republic of Korea) were used to measure the protein concentrations of the samples. Next, proteins (25 μg) was analyzed by SDS-PAGE using Precast 4–15% gels (Mini-PROTEAN^®^ TGX™ Precast Protein Gels, Bio-Rad, Hercules, CA, USA) and then transferred to nitrocellulose membrane (Bio-Rad). After blocking with 5% skim milk, membranes were incubated with different primary antibodies overnight at 4 °C. Rabbit anti-COL1α1, rabbit anti-GAPDH, rabbit anti-RUNX2, rabbit anti-Phospho NF-κB, rabbit anti-Phospho p38-MAPK, mouse anti-PCNA, rabbit anti-p21 (cell signaling), rabbit anti-OCN (Biorbyt, Cambridge, UK), mouse anti-β-actin and mouse anti-p16^ink4a^ (Santa Cruz Biotechnology, Dallas, TX, USA) were used as primary antibodies. Secondary horseradish peroxidase-conjugated anti-mouse and anti-rabbit antibodies were from Donkey (Jackson ImmunoResearch Europe Ltd., Cambridge, UK). Protein bands were visualized using Clarity ECL chemiluminescence substrate (Bio-Rad) with Uvitec Imager (UVItec, Cambridge, UK) and then quantified using ImageJ software (version 1.52a, Stuttgart, Germany).

### 4.10. ELISA Assay

Cell culture supernatants were collected after each treatment, centrifuged and stored at −80 °C until use for MCP-1 (RayBiotech Life, Inc., Peachtree Corners, GA, USA) and IL-8 (Invitrogen, Waltham, MA, USA) quantification by ELISA, according to the manufacturer’s protocol.

### 4.11. Alkaline Phosphatase (ALP) Assay

The ALP level in MSC and sMSC was measured in cell lysates using a commercial ALP Assay Kit from Abcam (cat# ab83369), following the manufacturer’s instructions. Briefly, cell lysates were quantified with Bradford assay and total protein (25 μg) was analyzed. Then, the production of p-nitrophenyl phosphate was determined by measuring the absorbance at 405 nm using a microplate reader.

### 4.12. Statistical Analysis

Data were analyzed and visualized using GraphPad Prism (version 7, San Diego, CA, USA). The values were presented as mean ± SD or frequency (%). Paired sample t-test was used for the analysis of RT-qPCR, ELISA and densitometric data. *p* value < 0.05 was considered a statistically significant difference.

## 5. Conclusions

In this study we showed that orthosilicic acid, vitamin K2, curcumin, polydatin and quercetin are safe bioactive compounds that synergistically act in promoting osteoblastic differentiation of MSC and inhibit the inflammatory phenotype associated with cellular senescence. Due to the crucial role of MSC in maintaining bone health, their combination in the nutraceutical BlastiMin Complex^®^ (Mivell, Italy) could represent a valuable dietary supplement in the prevention of, or as an adjuvant in, age-related bone loss. A clinical trial to test this hypothesis is currently underway.

## 6. Patents

European Patent application: EP22216685.2.

## Figures and Tables

**Figure 1 ijms-24-08820-f001:**
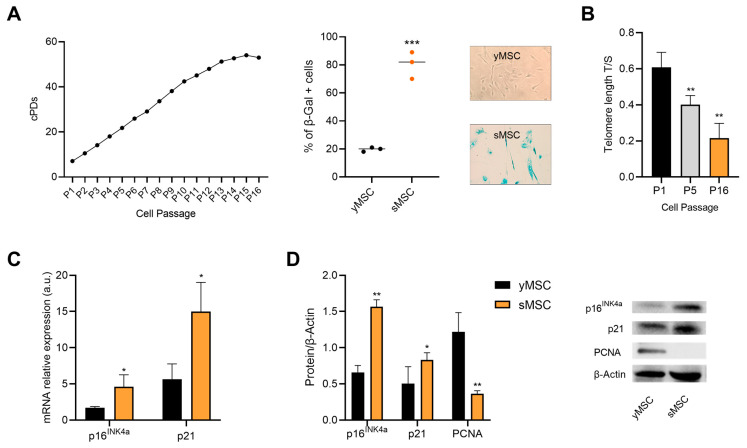
In vitro characterization of replicative senescent MSCs (sMSCs). (**A**) Characterization of sMSC by cumulative population doubling curve and (SA)-β-Gal activity. (**B**) Telomere length in bone marrow mesenchymal stromal cells, calculated according to 2^−ΔΔCt^ method using a single-copy reference DNA for normalization. (**C**) p16^ink4a^ and p21 mRNA expression. Data are reported as relative expression according to 2^−ΔΔCt^ method using IPO8 as housekeeping. (**D**) Representative Western blot analysis of p16^ink4a^, p21 and PCNA protein expression and densitometric analysis using β-Actin as loading control. The results are expressed as mean ±SD from three independent biological replicates. Paired *t* test, * *p* < 0.05, ** *p* < 0.01, *** *p* < 0.001 vs. yMSC.

**Figure 2 ijms-24-08820-f002:**
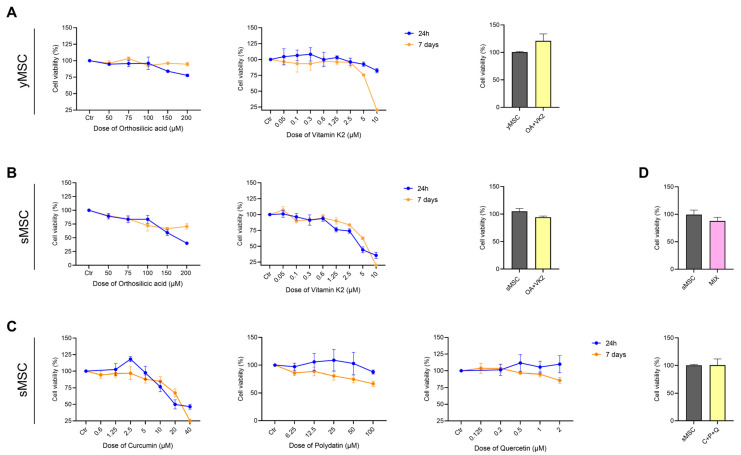
Dose-response curve of yMSC and sMSC to orthosilicic acid, vitamin K2, curcumin, polydatin, quercetin and their combinations. (**A**) yMSC and (**B**) sMSC were treated with different concentrations of orthosilicic acid (from 50 μM to 200 μM), vitamin K2 (from 0.05 μM to 10 μM), their combination (OA+VK2, 75 μM OA and 0.1 μM VK2) or with PEG-60 hydrogenated castor oil (vitamin K2 solvent) alone as a control for 24 h or 7 days. (**C**) sMSC were treated with different concentrations of curcumin (from 0.6 μM to 40 μM), polydatin (from 6.25 μM to 100 μM), quercetin (from 0.125 μM to 2 μM), their combination (C+P+Q, 1 μM CUR, 10 μM PD, 0.5 μM QCT) or with DMSO alone as a control for the indicated times. (**D**) sMSC were treated for 7 days with the combination of OA, VK2, CUR, PD and QCT (MIX) or with DMSO alone as a control. The MTT assay was used to assess cell viability upon treatment. The results are presented as a percentage of cell viability normalized to the viability of PEG-60 hydrogenated castor oil-treated (Ctr) for VK2 or DMSO-treated cells (Ctr)—for CUR, PD and QCT—and presented as mean value ±SD from three independent biological replicates.

**Figure 3 ijms-24-08820-f003:**
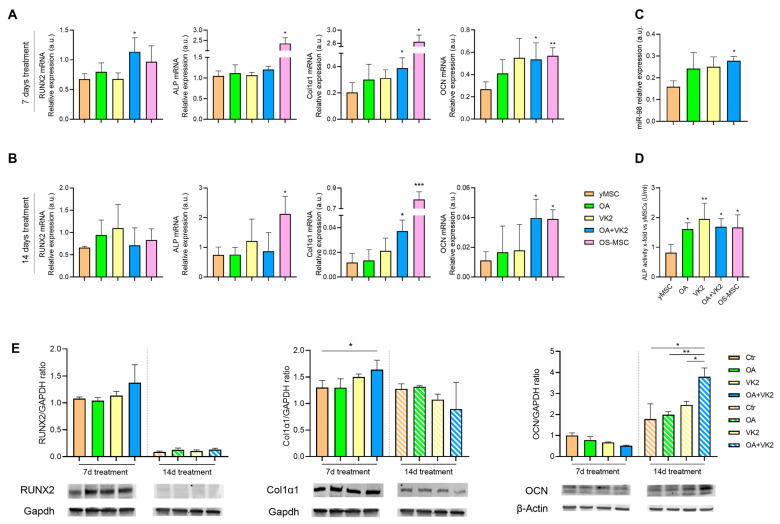
Orthosilicic acid and Vitamin K2 have an osteogenic effect on yMSC. yMSC were treated with OA (75 μM), VK2 (0.1 μM) and their combination (OA+VK2) and RUNX2, ALP, Col1α1 and OCN mRNA expression analyzed after 7 and 14 days of treatment (**A**,**B**). (**C**) miR-98 expression analysis in 7 days-treated yMSC. Data are reported as relative expression according to 2^−ΔΔCt^ method, using IPO8 or RNU44 as housekeeping. (**D**) ALP activity of 7 days treated with yMSC. Data are reported as fold change vs untreated yMSC. (**E**) Representative Western blot analysis showing RUNX2, Col1α1 and OCN protein expression, with Gapdh or β-actin as loading controls. The bands were quantified by ImageJ. All data are reported as fold change vs untreated yMSC. The results are expressed as mean ± SD from three independent biological replicates. Paired *t* test, * *p* < 0.05, ** *p* < 0.01, *** *p* < 0.001 vs. untreated yMSC.

**Figure 4 ijms-24-08820-f004:**
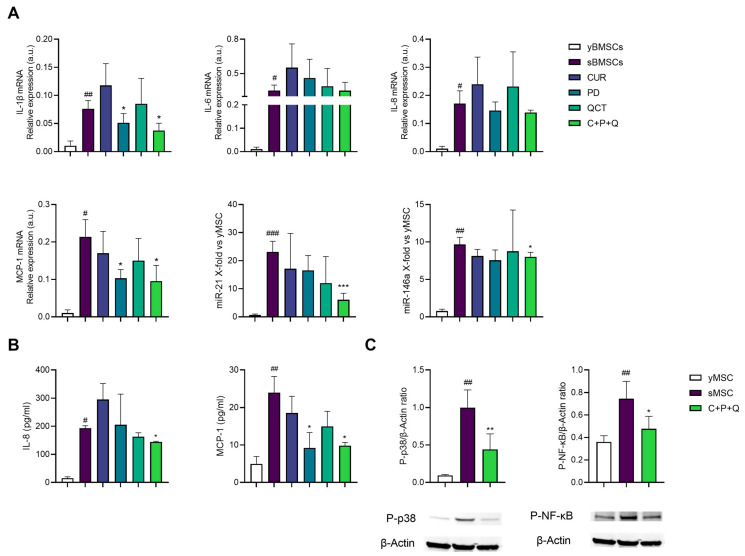
Curcumin, polydatin and quercetin exert an anti-inflammatory effect on sMSC. (**A**) IL-1β, IL-6, IL-8, MCP-1 mRNA and miR-21 and miR-146a expression of sMSC treated with CUR, PD and QCT for 24h. Data are reported as relative expression according to 2^−ΔΔCt^ method, using IPO8 and RNU44 as housekeeping. (**B**) Concentration (pg/mL) of IL-8 and MCP-1 in the conditioned medium of untreated and treated sMSC. Histograms represent the mean of three independent experiments ± SD. (**C**) Representative Western blot analysis showing P-p38 and P-NF-κB protein expression, using β-actin as loading control. The bands were quantified by ImageJ. The results are expressed as mean ± SD from three independent biological replicates. Paired *t* test, # *p* < 0.05, ## *p* < 0.01, ### *p* < 0.001 vs. untreated yMSC, * *p* < 0.05, ** *p* < 0.01, *** *p* <0.001 vs. untreated sMSC.

**Figure 5 ijms-24-08820-f005:**
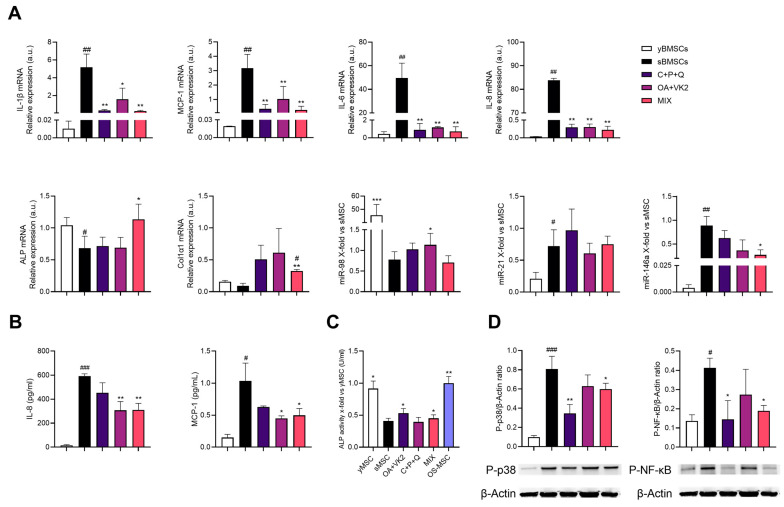
Anti-inflammatory and pro-osteogenic effect of OA+VK2, C+P+Q and their combination (MIX) in sMSC. (**A**) IL-1β, MCP-1, IL-6, IL-8, ALP, COL1α1 mRNA and miR-98, miR-21, miR-146a expression in sMSC treated with OA+VK2, C+P+Q and MIX for 7 days and yMSC (not differentiated). Data are reported as relative expression according to 2^−ΔΔCt^ and using IPO8 and RNU44 as housekeeping. (**B**) Concentration (pg/mL) of IL-8 and MCP-1 in conditioned media of yMSC, untreated and treated sMSC. (**C**) ALP activity in sMSC treated for 7 days with OA+VK2, C+P+Q and MIX. (**D**) Representative Western blot analysis of P-p38 and P-NF-κB protein expression with β-actin as loading control. The bands were quantified by ImageJ. The results are expressed as mean ± SD from three independent biological replicates. Paired *t* test, # *p* < 0.05, ## *p* < 0.01, ### *p* < 0.001 vs. untreated yMSC, * *p* < 0.05, ** *p* < 0.01, *** *p* < 0.001 vs. untreated sMSC.

**Figure 6 ijms-24-08820-f006:**
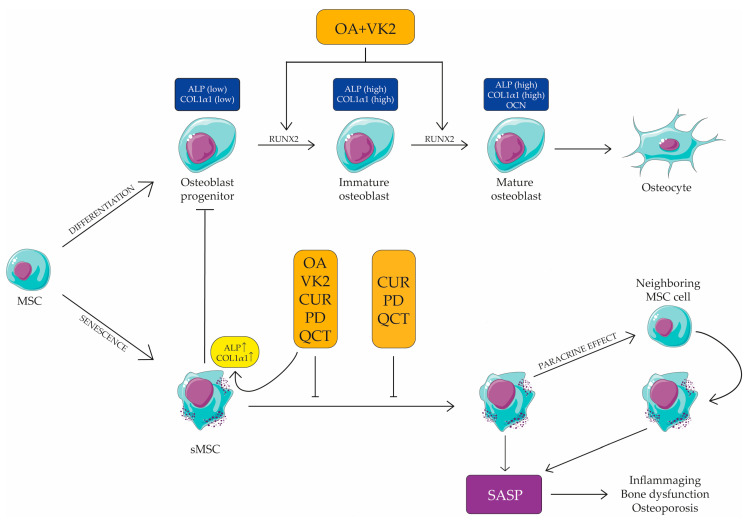
Schematic representation of orthosilicic acid (OA), vitamin K2 (VK2), curcumin (CUR), polydatin (PD) and quercetin (QCT) effects on MSC and sMSC. OA and VK2 combination up-regulated RUNX2, ALP and COL1α1 (upper part of the figure). CUR, PD and QCT combination inhibited SASP, which plays a role in low bone degeneration in aging. The association of the five compounds (OA, VK2, CUR, PD and QCT) and increased ALP and COL1α1 expression and inhibited SASP in sMSC. Overall, it can be suggested that reducing inflammation and promoting MSC (both young and senescent) differentiation by using these natural compounds might be a valid strategy to prevent bone loss during aging. This figure was created using the Servier Medical Art Commons Attribution 3.0 Unported Licence (http://smart.servier.com (accessed on 21 April 2023)).

**Table 1 ijms-24-08820-t001:** Synergism between the two active compounds orthosilicic acid and vitamin K2 on osteogenesis.

**yMSC—7 Days**	**RUNX2**		**ALP**		**Col1α1**		**OCN**	
	**Effect vs. yMSC (%)**	**Synergistic Effect ***	**Effect vs. yMSC (%)**	**Synergistic Effect ***	**Effect vs. yMSC (%)**	**Synergistic Effect ***	**Effect vs. yMSC (%)**	**Synergistic Effect ***
Orthosilicic acid	18	Y	6	Y	48	N	64	N
Vitamin K2	0		1		53		154	
OA+VK2	68		15		91		133	
**yMSC—14 days**	**RUNX2**		**ALP**		**Col1α1**		**OCN**	
	**Effect vs. yMSC (%)**	**Synergistic Effect ***	**Effect vs. yMSC (%)**	**Synergistic Effect ***	**Effect vs. yMSC (%)**	**Synergistic Effect ***	**Effect vs. yMSC (%)**	**Synergistic Effect ***
Orthosilicic acid	43	N	1	N	14	Y	49	Y
Vitamin K2	66		63		79		63	
OA+VK2	7		16		214		232	

Y = Yes; N = None. * The effect of orthosilicic acid and vitamin K2 taken together (OA+VK2) is greater than the sum of their separate effects at the same doses.

**Table 2 ijms-24-08820-t002:** Synergism between the two active compound combinations.

sMSC 7 Days	IL-1β		MCP-1		IL-6		IL-8		ALP		COL1α1	
	Effect vs. yMSC (%)	Synergistic effect *	Effect vs. yMSC (%)	Synergistic effect *	Effect vs. yMSC (%)	Synergistic effect *	Effect vs. yMSC (%)	Synergistic effect *	Effect vs. yMSC (%)	Synergistic effect *	Effect vs. yMSC (%)	Synergistic effect *
OA+VK2	70	N	67	N	97	N	99	N	11		623	
C+P+Q	94		88		98		99		10	Y	497	N
MIX	96		91		98		99		87		310	

(OA+VK2 and C+P+Q). Y = Yes; N = None. * The effect of OA+VK2 and C+P+Q taken together (MIX) is greater than the sum of their separate effects at the same doses.

## Data Availability

Data is contained within the article or Appendix A.

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
