# Peer review of "Pro-Osteogenic and Anti-Inflammatory Synergistic Effect of Orthosilicic Acid, Vitamin K2, Curcumin, Polydatin and Quercetin Combination in Young and Senescent Bone Marrow-Derived Mesenchymal Stromal Cells"

_ijms, 2023, doi:10.3390/ijms24108820_

Round 1

Reviewer 1 Report

In this manuscript, the authors investigated anti-inflammatory and pro-osteogenic effect of OA and VK2, two pro-osteogenic factors and of CUR, PD and QCT, three anti-inflammatory compounds in young and senescent MSC in vitro.

The authors found that OA and VK2 maybe have promote mineralizing effect on yMSC by measuring the main osteogenic markers Runx2, ALP, CoL1a1 and OCN. 

 By measured the expression of pro-inflammatory cytokines, authors found that anti-inflammatory compounds have anti-inflammatory activity on sMSC.

 Combination with two kinds of compounds together treated sMSC, the results shown that the compounds downregulated the expression of pro-inflammatory cytokines are upregulated ALP expression, the data suggested that these natural compounds have a synergistic effect for osteogenic differentiation of MSCs.

Based on this manuscript data, the authors suggested that these natural compounds have a potential role as a supplement to prevent or control the progression of age-related osteoporosis.

This study is interesting investigation; manuscript writing is well, and easy to read. The results will be beneficial to the aging and osteoporosis patients, looking forward the clinical trial results or in vivo results.

Comments:

In supplementary figure, authors displayed one Alizarin red mineralization assay figure of yMSC in osteogenic media. However, why authors did not show the mineralization assay with compounds?

In figure 3A, ALP mRNA expression did not see the synergistic effect, however, in table 1 shown "Y".

In figure 3A, Col1a1 look like have the synergistic effect, but in table 1 shown "N". is a writing mistake?

In figure 4, except IL-1β and mir-21 significantly showed the synergistic effect, other cytokines seemingly did not show the synergistic effect for three anti-inflammatory compounds.

Author Response

We are grateful to the Reviewers and the Editor for their suggestions which helped to further improve the quality of the revised manuscript. We are going to thoroughly discuss the points raised by the Reviewers. Changes in the manuscript are highlighted in red. We hope that the revised manuscript is suitable for publication in the International Journal of Molecular Sciences. 

We thank the reviewer for his/her recommendations.

- In supplementary figure, authors displayed one Alizarin red mineralization assay figure of yMSC in osteogenic media. However, why authors did not show the mineralization assay with compounds?

Thank you for this comment. In this study, we induced osteogenesis to test MSC differentiation potential with a specific and complete pro-osteogenic medium. However, since the experiments with the natural compounds (OA and VK2) have been carried out in MSC basal medium (non-osteogenic), we couldn’t appreciate a significant calcium deposition with Alizarin Red staining, but we demonstrated that they successfully promote osteogenesis via the up-regulation of the pro-osteogenic mediators.

-In figure 3A, ALP mRNA expression did not see the synergistic effect, however, in table 1 shown "Y".

Thank you for this comment. The asterisks shown in the figures are related to statistical analysis, whereas tables indicate synergisms. There are some cases in which synergism didn’t correlate with a statistically significant difference, such as the one you have indicated. 

-In figure 3A, Col1a1 look like have the synergistic effect, but in table 1 shown "N". is a writing mistake?

Thank you for your comment. Similarly to the previous comment, there are some cases in which synergism didn’t correlate with a statistically significant difference. In figure 3A, the expression of Col1a1 for OA+VK2 is statistically different from the MSC control, however, the sum of the single compounds is greater than the effect of the combination (Table 1), therefore there is no synergistic activity. 

-In figure 4, except IL-1β and mir-21 significantly showed the synergistic effect, other cytokines seemingly did not show the synergistic effect for three anti-inflammatory compounds.

Thank you for your comment. Unfortunately, in figure 4 we couldn’t find any synergistic activity, therefore we didn’t add a table. The asterisks in the figure are related to statistical analysis.

Reviewer 2 Report

The manuscript „Pro-osteogenic and anti-inflammatory synergistic effect of Orthosilicic Acid, Vitamin K2, Curcumin, Polydatin and Quercetin combination in young and senescent Bone Marrow-derived Mesenchymal Stromal Cells” has an interesting and actual subject of the research field.

The authors has tested the hypothesis that the combination of two pro-osteogenic factors Orthosilicic acid (OA) and Vitamin K2 (VK2), and three other an-ti-inflammatory compounds, Curcumin (CUR), Polydatin (PD) and Quercetin (QCT) – that mirror the nutraceutical BlastiMin Complex® (Mivell, Italy) – would be effective in promoting MSC os-teogenesis, even of replicative senescent cells (sMSCs), and inhibiting their pro-inflammatory phenotype in vitro.

The considered subject of this study is very interesting, and results showed that i) the association of OA and VK2 promoted MSC differ-entiation into osteoblasts, even when being cultured without other pro-differentiating factors and ii) CUR, PD and QCT exerted an anti-inflammatory effect on sMSCs and also synergised with OA and VK2 in promoting the expression of the pivotal osteogenic marker ALP in these cells. Overall, these data suggest a potential role of using a combination of all these natural compounds as a supplement to prevent or control the progression of age-related osteoporosis.

The methodology and procedures were correctly used, but in introduction some new relevant reference need be considered.

A “Conclusion” paragraph can be a little extended.

The figures are necessary but need a little polish (ex. Figs 4 and 5), to be more readable.

Author Response

We are grateful to the Reviewers and the Editor for their suggestions which helped to further improve the quality of the revised manuscript. We are going to thoroughly discuss the points raised by the Reviewers. Changes in the manuscript are highlighted in red. We hope that the revised manuscript is suitable for publication in the International Journal of Molecular Sciences.

We thank the reviewer for his/her recommendations.

1-The methodology and procedures were correctly used, but in introduction some new relevant reference need be considered.

Thank you for this suggestion. In the introduction, we included some new references regarding other applications of Curcumin and Polydatin (lines 94-96).

2-A “Conclusion” paragraph can be a little extended.

Thank you for this comment. A Conclusion paragraph has now been added.

3-The figures are necessary but need a little polish (ex. Figs 4 and 5), to be more readable.

Thank you for this suggestion. We revised Figures 4 and 5, to better clarify the graphs and legends. We changed the cytokines release graphs’ colours and moved some panels in both figures, with a minor change in panel letters (Figure 5). We changed the main text, accordingly (lines 225, 226, 227). We also changed the Figure 3D graph colour to be thorough (we added those figures on the review tab of the revised manuscript).   

Reviewer 3 Report

This manuscript is research on “Pro-osteogenic and anti-inflammatory synergistic effect of Orthosilicic Acid, Vitamin K2, Curcumin, Polydatin and Quercetincombination in young and senescent Bone Marrow-derived Mesenchymal Stromal Cells”.  the research is quite impressive, but unacceptable without minor revision. comments are attached as a file

 Minor editing of English language required

Author Response

We are grateful to the Reviewers and the Editor for their suggestions which helped to further improve the quality of the revised manuscript. We are going to thoroughly discuss the points raised by the Reviewers. Changes in the manuscript are highlighted in red. We hope that the revised manuscript is suitable for publication in the International Journal of Molecular Sciences.

We thank the reviewer for his/her recommendations.

1- In the abstract, please highlight a brief summary of your % cell viability values and add your approximate yield analysis values in cell repair.             
Thank you for this suggestion. We highlighted a sentence regarding the non-cytotoxicity of the compounds tested in the abstract (line 28), however due to the limited amount of words available in the abstract we couldn’t describe extensively all the viability percentages. These aspects are more deeply described in the results section.           
The focus of this study was on senescent MSC differentiation potential and inflammatory phenotype, therefore we didn’t investigate cellular repair.

2- In the material and method section, please cite all experimental procedures with literature.  
Thank you for this suggestion. We apologise for not having cited several protocol references. We have now cited 3 papers regarding Alizarin Red Staining (line 348), cell viability assay (line 382) and telomere shortening PCR protocol (line 424).

3- In the % cell viability analysis findings, why the concentration values of quercetin, curcumin and K2 were in equal range were not compared. Comparisons made at equal concentrations will be more meaningful. Adding a comparison table will be more understandable.     
Thank you for your comment. In paragraph 2.2 we showed all our cell viability results obtained after treating the cells with different compound concentrations. The diverse ranges and orders of magnitude depended on several literature references we mentioned throughout the paper.  The concentrations selected for subsequent experiments were those that gave the higher cell viability (>70%, after 24h and 7 days of treatment), notwithstanding that in most cases cell viability is extremely high in many of the concentrations tested (i.e. polydatin, quercetin). We believe that a comparison table would be redundant since we also provided column graphs for compound combinations viability on the right-hand side of the figure.           

4- The optimum recommendation of these concentrations and cell viability data should be emphasized more clearly in the recommendation section of the conclusion.    
Thank you for this suggestion. We included a sentence regarding natural compound concentrations’ recommendations in the conclusion paragraph.              

5- Please add a conclusion paragraph after the discussion section.       
Thank you for your suggestion, we included a conclusion paragraph in the manuscript (paragraph 4).  

6- In the introduction section, the literature findings of curcumin, polydatin, and quercetin on damaged cell repair should be given in more detail. However, it would be more useful to comment on the effectiveness of combination uses later. The following research will help. “Biocidal Activity of Bone Cements Containing Curcumin and Pegylated Quaternary Polyethylenimine T Eren, G Baysal, F Doğan Journal of Polymers and the Environment 28, 2469-2480”       
Thank you for your suggestion. In this study, we focused on the effects of Orthosilicic acid, vitamin K2, curcumin, polydatin and quercetin in osteoblastic differentiation of MSC, with particular attention to their relevance in age-related bone loss. We didn’t explore damaged cell repair. However, we believe that implementing our literature references with curcumin and polydatin applications in bone would be helpful to underline better the importance that these compounds might have as dietary supplements (lines 93-95). 

7- Minor grammatical errors in English need to be corrected. Thank you for your comment. We corrected some typos and grammatical errors, accordingly: lines 22, 45, 106, 155, 156, 172, 173, 262, 283, 308, 313, 423, 433, 434.